# Utility of the Surgical Toggle-Pin Suture Technique for Left Displacement of the Abomasum in Pregnant Cattle

**DOI:** 10.3390/ani15182716

**Published:** 2025-09-16

**Authors:** Hideo Iso, Fumikazu Uchiyama, Kouki Yamashita, Rina Kuba, Takeshi Tsuka

**Affiliations:** 1Iso Veterinary Service, 451–14 Shimakata, Nasushiobara 329-3152, Japan; isovet@mh.point.ne.jp (H.I.); iso@vc-iso.co.jp (F.U.); yamashita001a@gmail.com (K.Y.); rina.fairyfair@gmail.com (R.K.); 2Clinical Veterinary Sciences, Joint Department of Veterinary Medicine, Faculty of Agriculture, Tottori University, 4-101, Koyama-Minami, Tottori 680-8553, Japan

**Keywords:** eyeleteer, Holstein cow, left abomasal displacement, pregnant, toggle-pin suture

## Abstract

In pregnant cows with left displacement of the abomasum, the affected abomasum is difficult to correct to its normal position with left-flank abomasopexy by a physically small surgeon. The surgical toggle-pin suture technique can compensate for the technical disadvantage of left-flank abomasopexy. Additionally, this technique can overcome the technical problem of the conventional (blind) toggle-pin suture technique, which forces treated cows to be positioned in dorsal recumbency, leading to postoperative abortion. This technique can contribute to reducing surgical invasiveness and shortening surgical times. Thus, this technique is considered a new, moderate surgical alternative for repositioning the left displacement of the abomasum in pregnant cows.

## 1. Introduction

Abomasal dislocation is a common periparturient condition in cows; approximately 90% of left displacement of the abomasum (LDA) occurs within the first six weeks after calving [1,2,3,4,5,6]. However, 2–10% of cows experience LDA during their mid-to-late pregnancy, predominantly in the last three weeks before calving [2,5,6,7,8]. In pregnant cows with LDA, a gravid uterus is considered an etiological factor, allowing dorsal displacement of the rumen and resulting in the creation of an anatomical void for the abomasum to move [1,7,9,10]. Left-displaced abomasum associated with the rearrangement of the abdominal viscera is considered to correspond to a semi-displacement of the abomasum, as the intra-abdominal condition occurs frequently during late pregnancy [7]. Mechanical effects from the gravid uterus do not always lead to LDA when occurring before the sixth month of pregnancy [2,5]. Common surgical techniques used to correct LDAs during pregnancy include left-flank abomasopexy [5,11,12,13]. Right-flank omentopexy is not commonly chosen as a surgical option because of the difficulty in repositioning the dislocated abomasum within the left abdominal space via the right-flank opening [14].

Previously, affected pregnant cows were treated with a rolling technique [15]. However, this technique can lead to a critical risk of inducing torsion of the gravid uterus when performed on pregnant cows at 6–8 months of gestation [4,15,16]. The toggle-pin suture (TPS) technique is also a minimally invasive technique in which the cow is positioned in dorsal recumbency while rolling clockwise during the procedure; this is referred to as the blind TPS technique [17,18,19]. This technique seems unsuitable for LDA correction in pregnant cows for the same reasons as the rolling technique [8,15,16].

Recently, one-step laparoscopic abomasopexy, also known as the laparoscopy-assisted TPS technique [11,18,20], has been effectively used for the surgical repositioning of the left-displaced abomasum during late pregnancy [21,22]. Briefly, the first step of this technique is a laparoscopy-guided procedure to insert the rod of a TPS device into the abomasal lumen through a cannula introduced initially via a trocar, followed by the escape of intraluminal gas [22]. The second step is a blind procedure to fix the suture part of the TPS device to the ventral abdominal wall using a 130 cm long needle [22]. In a previous study, this surgical procedure was successfully performed in 15 affected cows without disturbing the gravid uterus in any of them [22]. However, widespread adoption of this technique is limited because the endoscopic equipment required is expensive, and the surgeon requires a high level of skill. Thus, we designed a new TPS technique combined with left-flank abomasopexy, inspired by the surgical concept of one-step laparoscopic abomasopexy, referred to as the surgical TPS (sTPS) technique in the present report.

The purpose of the present study was to evaluate the therapeutic efficacy of sTPS for LDA during pregnancy, and to discuss the clinical applicability of the present technique in relation to conventional TPS and surgical techniques for pregnant cows with LDA.

## 2. Materials and Methods

### 2.1. Animals

The subjects were five Holstein milking cows reared on five dairy farms. A common clinical sign in these animals was a sudden decrease in or total loss of appetite during late pregnancy. Left-displaced abomasum was diagnosed based on an auscultation examination with percussion over the left intercostal and flank areas and hearing a ping sound. The animals were treated with sTPS between January 2020 and August 2022.

### 2.2. Surgical Procedures and Setting of Surgical Steps in sTPS

sTPS was performed with the animal in the standing position and sedated with an intramuscular injection of xylazine hydrochloride (0.05 mg/kg; Rompun; Elanco Japan, Minato-ku, Tokyo, Japan). The skin was shaved and disinfected at the ventral area of the left flank along the thirteenth rib and the midline area of the ventral abdomen more cranially than the mammary gland. Local anesthesia was induced by percutaneous injection of procaine hydrochloride (30 mL; Adsan; Riken Vets Pharma Inc., Ogose-cho, Saitama, Japan) diffused within the surgical areas, as infiltration anesthesia.

Step 1 (Opening in the abdominal cavity): A 15 cm-long skin incision was made in the ventral area of the left flank 2 to 3 cm caudal to the left thirteenth rib. The middle of the incision was at the height of the costochondral junction of the rib (Figure 1A). The surface of the left-displaced abomasum was detected directly below the surgical opening created by the incision of the muscular layers and the underlying peritoneal membrane (Figure 1B). No gravid uterus was observed through the surgical opening.

Step 2 (Abomasal centesis and repositioning): Under macroscopic observation, a cannula (5 mm in outer diameter and 15 cm in length; self-made) was initially introduced into the dislocated abomasum by the centesis procedure, followed by insertion of the rod of a TPS device (rod size: 5 mm × 30 mm; suture length: 70 cm [USP3]; BS Medical Co., Ltd., Bunkyo-ku, Tokyo, Japan) into the abomasal lumen (Figure 1C). The introduced cannula was removed immediately after the TPS setting. Subsequently, a second introduction of the same cannula was quickly conducted at different parts of the abomasum, followed by insertion of the rod part of the TPS device. During these two TPS setting procedures, intraluminal gas escaped, resulting in the descent of the upwardly displaced abomasum. The abomasum was finally pushed down toward the ventral region of the abdominal cavity by the operator after the removal of two cannula, following the natural descent of the abomasum after gas evacuation.

Step 3 (Fixation of the abomasum to the abdominal wall): A suture of the TPS device, which had already been set within the abomasal lumen, was threaded through a hole at the sharp tip of a commercial eyeleteer (1 mm tip and 16.5 cm total length; Fujiwara Sangyo Co., Ltd., Miki-city, Hyogo, Japan) D and Figure 2A). The surgeon inserted his arm into the abdominal cavity through the surgical opening while tightly gripping the setting device (Figure 1E). The surgeon advanced his hand while gripping the device to the deepest area, where the inner surface of the ventral abdominal wall could be felt. The assistant veterinarian identified an adequate fixation location while monitoring the movement of the skin surface of the ventral abdomen synchronized with the manual pushing down of the abdominal cavity by the surgeon (Figure 1F). With the help of an assistant, the eyeleteer tip, in which a TPS was set, was released and then stuck and protruded outside at the midline of the ventral abdominal wall cranial to the udder and caudal to the xiphoid process (Figure 2B). The suture of the TPS device was picked up by an assistant veterinarian. This procedure was repeated twice. The assistant veterinarian tied the two sutures of the two TPS devices together on the ventral abdominal skin while pulling carefully so that the abomasum could come into sufficient contact with the inner surface of the abdominal wall.

Step 4 (Surgical wound closure): The surgeon stitched the peritoneum and muscular layers including the abdominal oblique and rectus muscles using an absorbable suture material (Opepolix, USP-3, Alfresa Pharma Co., Osaka-city, Osaka, Japan), followed by skin suturing using a nylon suture material (Suprylon USP5, Vömel, Gronberg, Germany).

The animals were postoperatively treated with a 3-day intramuscular administration of antibiotics (450,000 units/100 kg; penicillin G procaine; Riken Vets Pharma Inc., Ogose-cho, Saitama, Japan). Skin suturing was removed 7 days after surgery. Simultaneously, the tied part of the TPS to fix the abomasum and the ventral abdominal wall was also cut. The TPS devices were observed in the feces 1 to 2 days after the sutures were cut.

### 2.3. Clinical Data

The clinical data included individual records of operation days (age, parity, and gestation day) and postoperative records (the interval between operation and calving, and between operation and culling). The interval between calving after surgery and the next calving, referred to as the calving interval, was also recorded.

## 3. Results

Mean gestation period was 267.8 days when treated with sTPS in five cases (Table 1). It took 1.6 ± 0.6 and 3.4 ± 1.1 min to perform steps 2 and 3, respectively, including fixation of the displaced abomasum to the ventral abdominal wall using a TPS device (Table 2). The surgical times in steps 2 and 3 tended to be shorter than those in steps 1 and 4, including the operations to open and close the left abdominal wall. The total surgical time was 17.2 ± 2.3 min. The treated cows showed gradual improvements in appetite, resulting in a return to the normal state four days after surgery. The intervals between operation and calving were 16.8 ± 6.0 days (Table 1). During the lactation periods after calving following surgery, a calving interval of 430.6 ± 14.2 days was found, ranging between 412 and 443 days. Within the follow-up period, there was no recurrence of LDA in any of the five animals treated with sTPS. Two of the five cows treated with sTPS were culled 902 and 1525 days after surgery.

## 4. Discussion

In the present study, the mean surgical time required for sTPS was 17 min, as the time taken for the surgical process of steps 1 to 4. The surgical time required for the entire closed suturing technique using a bar suture or TPS device is 10–15 min [3,18,23,24,25,26]. The sTPS technique is considered less surgically invasive when performed in the standing position, although it requires slightly longer surgical times than the blind TPS technique. The surgical times required for sTPS were almost identical to or slightly longer than those required for the laparoscopy-assisted TPS technique (15–40 min and 15–20 min, respectively), which were performed during the late pregnancy and lactation periods, respectively [21,22,27].

In terms of the surgical time required for each step of the sTPS techniques to correct the LDAs in the present cases, the length of surgical time required for the opening and closure of the abdominal cavity (steps 1 and 4, respectively) appeared to be associated with the size of the surgical wounds. In step 2 of the sTPS, the abomasum was directly observed via the left flank opening, contributing to the quick intraluminal gas evacuation and TPS setting. sTPS also helps identify abomasal wall perforations and adhesive changes, similar to left-flank abomasopexy, which are common surgical options for correcting LDAs during late pregnancy [14]. This may have contributed to the shorter surgical time in step 2. Step 3 of the sTPS is considered a more difficult procedure, but it took approximately 3 min to complete. In step 3 of the sTPS, the surgeon could not confirm the fixation of the abomasum in the functional position during surgery. Thus, reduction in surgical time during step 3 appeared to depend on the assistant veterinarian’s excellent skill in supporting the procedure.

The recommended interval for TPS fixation is not well known, although it has been reported to be 2–4 weeks after surgery [17,19,28]. Additionally, the postoperative decision to permanently maintain TPS fixation without suture cutting is made because TPS removal rarely causes LDA recurrence, although it is performed after the recommended TPS fixation intervals [17]. TPS fixation is clinically removed between postoperative days 14 and 21, according to previous studies [4,29]. In our sTPS procedure, the 7-day postoperative interval for TPS fixation removal might be too short based on the healing reaction, as the fixation time required for stable fibrous adhesion of the corrected abomasum to the abdominal wall is estimated to be between 10 and 14 days [18,20]. Thus, the interval between surgery and removal of the TPS in sTPS should be extended to decrease the risk of recurrence. In sTPS, a TPS device is unlikely to be the source of infection within the abdomen, because this device was eliminated from the body after cutting its suture part. On the other hand, in left-flank abomasopexy, the suture materials are retained at the fixation points between the abomasum and abdominal walls after cutting. The retained sutures can lead to postoperative infection, causing abscess formation and penetration of the abomasal lumen [28].

In the present report, an intramuscular injection of xylazine (0.05 mg/kg) was used for sedation during the sTPS. Sedation for laparotomic abomasal surgery in the standing position can be achieved by intravenous injections of xylazine at a dosage of 0.02–0.05 mg/kg [4,26,30,31,32,33,34,35]. The cows were sedated with intravenous injections of xylazine (0.05–0.1 mg/kg) when treated with the blind TPS technique [3,26]. However, this dosage of xylazine (0.05 mg/kg, intravenously) can cause abortion in pregnant cows, as it is known to account for 3.7% of the abortion rate [31]. Care should be taken when using α2-adrenergic agonists in pregnant cows, because the drug can stimulate myometrial activity of the gravid uterus (oxytocin-like effect), causing abortion or precipitating premature parturition, and inducing a reduction in uterine blood flow, causing fetal death [30,31,32,33,34]. This is supported by an in vitro study using a bovine uterus, in which xylazine enhanced contractility at 30–60 days [34]. The dose-dependent pharmacological effects of xylazine differ between the intramuscular and intravenous routes; the adverse effects of this drug, such as reduced uterine blood flow, seem to be milder when administered intramuscularly than when administered intravenously [32]. Furthermore, the sedative effect of an intravenous dose of 0.02 mg/kg xylazine was found to be similar to that of an intramuscular dose of 0.05 mg/kg of the same sedative [35]. Although our xylazine dosage might be safe for pregnant cows, sedation should not be used to perform sTPS in pregnant LDA animals to reduce the risk of abortion as much as possible. Sedation may not always be required for sTPS because it is a minimally invasive surgical technique. Additionally, proper administration of local anesthesia can replace the use of xylazine in sTPS [28].

The recovery and success rates in cows treated with bar sutures or blind TPS techniques have previously been reported to be between 71% and 93% [3,6,18,19,23,36], with mortality rates of 6–13% [23,36]. Another retrospective study revealed that 8.7% and 10.8% of treated cows died 14 and 60 days after blind TPS techniques, respectively, owing to various causes, including intra- and postoperative complications [25]. Such complications can occur partly depending on the blind setting of the TPS device in the abomasal lumen, allowing accidental failure of the intraluminal setting, TPS misplacement, and laceration of the adjacent visceras [18,19]. Additionally, the blind procedure can occasionally result in multiple trials in the setting; multiple procedures for intraluminal device insertion (>three times) seem to correlate with an increased risk of culling [19]. The sTPS technique allows for accurate and rapid introduction of the rod of the TPS device into the abomasal lumen twice during the procedure. The two pieces of the TPS device should be set as quickly as possible before the abomasum deflates [12].

The survival rates of cows treated with the blind TPS technique have varied among previous studies, with values of 84% and 75% on postoperative days 14 and 60, respectively, in one report [25]; and 84–87% on postoperative day 120 in other reports [3,18,37]. Grymer reported that 90% of treated cows survived 50 days postoperatively, with a one-year survival rate of 60% [37]. The median survival time after treatment with laparoscopy-assisted TPS was 717 days, ranging from 52 to 1706 days [4]. Pregnant LDA cows treated with this technique survived for a mean of 415 days, ranging from 83 to 962 days, with a one-year survival rate of 60% [22]. Survival times in previous cases were shorter than those in cows treated with sTPS. Differences in survival rates seem to depend on the degree of culling pressure on herd owners [10,25]. For cows treated using the blind TPS technique, the culling risk increases approximately 10 times for randomly selected herd mates [38]. The owner’s culling pressure may be lower in cows requiring sTPS for LDA correction than in cows treated with the blind TPS technique.

Follow-up without surgical intervention may be preferable in affected cows with clinical appearances associated with LDAs during late pregnancy, because slight abomasal displacements may occur normally during this period, a condition known as semi-displacement of the abomasum [7,9]. However, all five cows exhibited a loss of appetite, leading to the decision to proceed with surgical intervention. Long-term nutritional deficits associated with a severe decrease or loss of appetite during pregnancy can contribute to poor milk production and reproductive performance after calving [22]. Calving intervals tended to be slightly extended in the five cows treated with sTPS; however, there were no reproductive records, including the intervals between calving and first service, number of artificial inseminations, and days open, which are the common factors associated with this value [39,40]. These values were within or slightly beyond the average for lactating cows, which reportedly range from 383 to 435 days in the United States, Canada, Ireland, and the Netherlands [39,40]. The involvement of LDA may have a negative effect on the calving interval, which is estimated to be 58 days longer than that of unaffected cows [4,41]. Prolonged calving intervals are possibly caused by nutritional disturbances associated with the involvement of LDA, subsequently inducing poor reproductive performance, such as a delay in the first service after calving [4,22,41]. However, the calving intervals of the five animals included in the present study were slightly shorter than the average values recorded at the five farms (A to E) where they were kept (441,459, 444, 451, and 423 days in farms A, B, C, D, and E, respectively).

Left-flank abomasopexy is commonly applicable for surgical correction of LDA occurring during pregnancy [5,11,12,13]. However, a physically small surgeon with very short arms may be more unsuitable for performing left-flank abomasopexy during pregnancy, because LDA correction is challenging with this technique for anyone [11]. Thus, we designed the procedure of sTPS to address the problem and reduce surgical invasiveness for the pregnant, treated animals. However, significant technical improvements are required to further enhance the clinical application of sTPS. In this technique, the surgical opening to approach the left-replaced abomasum should be made at the height of the costochondral junction of the left thirteenth rib, as the technique may be performed by a short-armed or not very large surgeon, similar to the approach of left-flank abomasopexy that such persons perform [14]. However, the sTPS procedure appears ergonomically uncomfortable to perform because the surgeon is consistently bent over or squatting during the surgery. In five cases, the operator could detect the greater curvature of the abomasum, as the recommended part to secure with the abdominal walls, through the surgical opening. However, a surgical approach through this opening may not always allow observation of the greater curvature when it is displaced upward beyond the opening. Additionally, this lower position of the surgical opening can cause accidental leakage of intraluminal fluid contents via a cannula when introduced into the abomasal lumen in step 2 of sTPS, although this problem did not occur in any of the five present cases. Abomasal atony causes excess intraluminal accumulation of gas, which commonly floats in the dorsal region of the abomasal lumen [14]. The layers of fluid or solid ingesta and gas cap are separated ventrally and dorsally, respectively, within the displaced abomasum [1,14]. Thus, the upper part of the displaced abomasum appears to be an adequate location for setting up the cannula to allow the effective evacuation of intraluminal gas. The development of an alternative device, such as a long applicator in the laparoscopy-assisted TPS technique, is required for adequate centesis of the displaced abomasum [22].

## 5. Conclusions

The sTPS technique introduced in the present study can lead to greater surgical invasiveness than the conventional (blind) TPS technique because a skin incision followed by opening of the abdominal cavity is required to set a TPS device to correct the left-displaced abomasum. However, this surgical procedure contributes to abomasal repositioning while keeping the treated animals standing, whereas the conventional TPS procedure requires the animal to be placed in dorsal recumbency. When using a TPS device on abomasal repositioning for pregnant cows, the restraint of dorsal recumbency can lead to greater stress for the treated cows than the creation of a small surgical incision. Thus, the sTPS technique is more suitable for repositioning the left-displaced abomasum during gestation. To further increase its applicability, it is necessary to improve the surgical device to make the surgical opening in a higher location than the present procedure of sTPS, contributing to surgeon comfort and prevention of contingencies during surgery.

## Figures and Tables

**Figure 1 animals-15-02716-f001:**
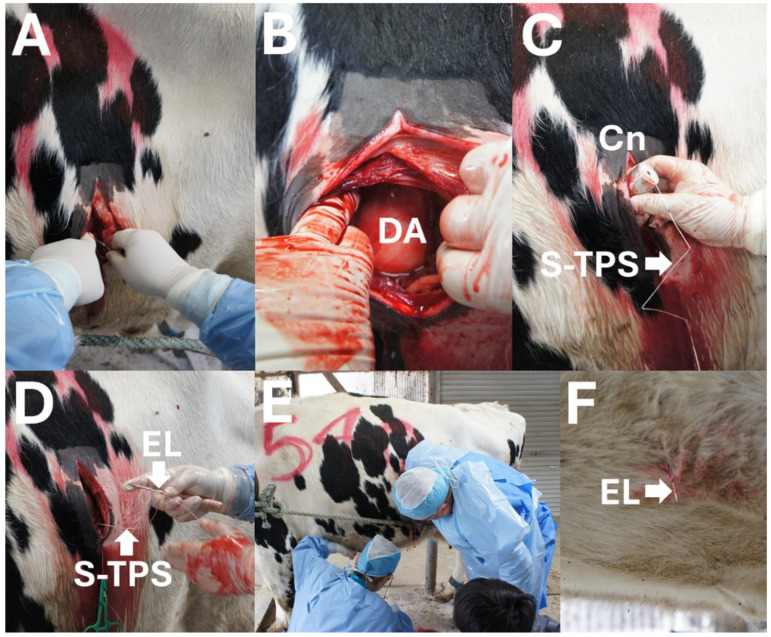
Intraoperative photos of surgical toggle-pin suture technique. (**A**) Skin incision is made at the ventral area of the left flank. (**B**) Left-displaced abomasum (DA) is macroscopically detected through the surgical opening. (**C**) A toggle-pin suture device is inserted into the lumen of the DA through a cannula (Cn) while the suture part of the toggle-pin suture device (S-TPS) is held outside. (**D**) An S-TPS is set at the sharp tip of an eyeleteer (EL) manually gripped by the surgeon (I.H.). (**E**) The surgeon looks for an adequate place to fix the abomasum with the help of an assistant (R.K.). (**F**) An EL tip, in which an S-TPS is set, is stuck and protrudes outside at the ventral abdominal wall.

**Figure 2 animals-15-02716-f002:**
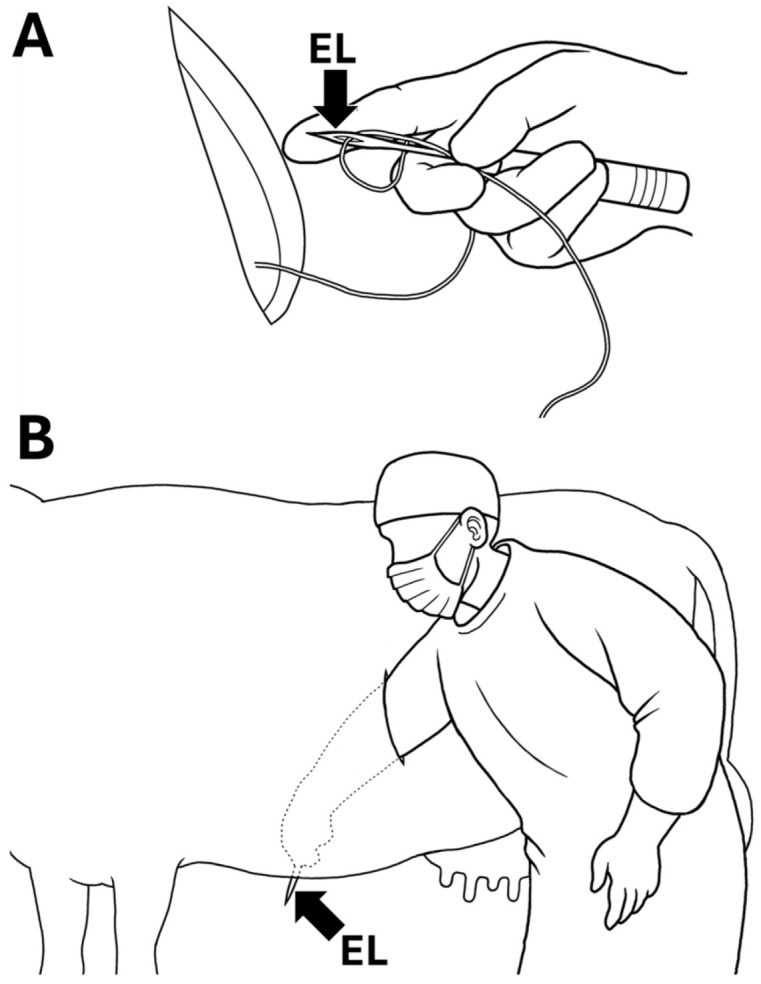
Diagram of surgical toggle-pin suture technique. (**A**) A suture of toggle-pin suture device is threaded through a hole at the sharp tip of an eyeleteer (EL). The surgeon holds the suture in the setting place using his finger to prevent it from slipping out of the hole. (**B**) The surgeon extends his arm and inserts it through the surgical opening toward the ventral abdominal wall, and then pierces the wall with an eyeleteer tip, in which a suture of toggle-pin suture device is set.

**Table 1 animals-15-02716-t001:** Clinical records in operation days and postoperative data in the present cases.

Cow No. ^a^	Farm	Age (Day) at Operation	Parity at Operation	Gestation Day at Operation	Postoperative Day to Calving	Calving Interval	Days Between Operation and Culling
sTPS1	A	1237	2	269	16	419	1525
sTPS2	B	1130	2	267	18	443	902
sTPS3	C	2567	4	263	22	437	Alive ^b^
sTPS4	D	2638	3	262	21	442	Alive ^b^
sTPS5	E	982	1	278	7	412	Alive ^b^
Mean (SD)		1710.8 (819.4)	2.4 (1.1)	267.8 (6.4)	16.8 (6.0)	430.6 (14.2)	

^a^ sTPS shows the cows treated with the surgical toggle-pin suture technique. ^b^ There was no culling record by 31 December 2024.

**Table 2 animals-15-02716-t002:** Surgical times (min) required for the surgical toggle-pin suture technique (each of the four steps and in total) in pregnant cows with left displacement of the abomasum.

Cow No. ^a^	Step 1	Step 2	Step 3	Step 4	Total
sTPS1	5	1	2	7	15
sTPS2	4	2	3	8	17
sTPS3	6	1	4	9	20
sTPS4	4	2	3	6	15
sTPS5	4	3	5	8	19
Mean (SD)	4.6 (0.9)	1.6 (0.6)	3.4 (1.1)	7.6 (1.1)	17.2 (2.3)

^a^ sTPS shows the cows treated with the surgical toggle-pin suture technique.

## Data Availability

The data presented in this study are available on request from the corresponding author.

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
