# Peer review of "Utility of the Surgical Toggle-Pin Suture Technique for Left Displacement of the Abomasum in Pregnant Cattle"

_animals, 2025, doi:10.3390/ani15182716_

Round 1
Reviewer 1 Report (Previous Reviewer 1)
Comments and Suggestions for Authors
The manuscript has substantially improved compared to the previous version. The authors have addressed my suggestions, including the discussion of the study’s limitations. In my opinion, the manuscript is now suitable for publication.
Author Response
Comments and Suggestions for Authors
The manuscript has substantially improved compared to the previous version. The authors have addressed my suggestions, including the discussion of the study’s limitations. In my opinion, the manuscript is now suitable for publication.
Response: Thank you for your kind suggestion for the improvement of our paper. In the revised version, the corrected parts are highlighted with yellow boxes according to suggestions from Reviewers 1 and 2.
Reviewer 2 Report (New Reviewer)
Comments and Suggestions for Authors
Summary
The article describes the surgical treatment and subsequent follow-up of five cases of LDA in pregnant cows during the last month of pregnancy. Although similar clinical cases have been described in individual cows, to our knowledge, there are no clinical case series of LDA in cows at advanced stages of pregnancy. Furthermore, the proposed surgical technique will be useful for surgeons of small stature. In addition, the results are exceptionally good. For all this we congratulate the authors.
The authors justify the need for the new technique because being one-step laparoscopic abomasopexy is a highly effective method for correcting left displacement of the abomasum (LDA) in pregnant cows, it is not accessible to all veterinarians because of the need for expensive equipment and substantial experience.
In addition, the study argues that the toggle-pin suture (TPS) technique is also not appropriate for advanced pregnant cows due to the risk of uterine torsion.
The introduction and discussion focus on evaluating the efficacy of sTPS in relation to conventional TPS and laparoscopy-assisted TPS.
But, the most significant weakness observed in this work is that the technique recommended in all textbooks, and papers (Your reference number 15: Niehaus, A.J. Surgical management of abomasal disease) for LDA in pregnant cows is the left flank abomasopexy or Utrecht method. In the same books it is said that if the surgeon is small, the incision simply needs to be extended downward.
However, this technique is only mentioned in the introduction and is not addressed in the discussion. In our opinion, sTPS is a left-flank abomasopexy in which the incision is made more ventrally, to facilitate the work of smaller veterinarians, and in which the abomasopexy is performed with a toggle-pin suture instead of a conventional suture. Therefore, both the introduction and discussion should address left-flank abomasopexy in greater detail.
On the other hand, the advantages and disadvantages of the traditional abomasal suture used in the Utrecht method versus the toggle-pin suture used in this paper should be discussed.
If you could add those aspects, the paper would undoubtedly improve.
Major issues
Introduction
-Line 41 and 42: Two-flank abomasopexy is not a common surgical technique, the 3 cited cases are of individual problematic animals with concomitant problems.
-Line 42: Cite the original authors of the left flank abomasopexy (Lagerwey E, and Numans SR. De operatieve behandeling van de lebmaagdislocatie bij het rund volgens de ‘Utrechtse methode’. TijdschrDiergeneeskd 1968; 93: 366-76.) Since citation number 13 is not very appropriate for a paper as it is a workshop.
M&M
-Lines 86 to 88: Given that Figure 1 does not show the entire cow, that the ventral area of the left flank is a very large area, and that the novelty of the technique is the incision site, can you specify exactly where the incision is located: how many cm caudal to rib 13, and whether the dorsal edge of the incision is at the level of the costochondral junction of said rib or is it located higher? A lateral photograph of only the patient with the incision made would be very illustrative.
-Lines 92 to 102: In many cases, the LDA is well above the incision line. In the Utrecht technique, abomasopexy is performed on the greater curvature of the abomasum, in the most dorsal part of it when it is displaced, but with a 15 cm rigid cannula and a low incision. How is the greater curvature accessed? Detail precisely in which parts of the abomasum the cannula is introduced.
-Lines 94-95: Explain the materials used to make the rod and suture.
-Lines 109 a 112: Determine the exact anatomical location of the fixation area on the patient's abdominal wall.
-Lines 119 to 121: Figure 1 B shows at least two muscular layers. But the text says "muscular layer." Can you specify which muscles and fascia are sutured and how?
-Lines 119-122: Specify the suture material since neither Opepolix nor Suprylon are available on the Internet.
-Line 136, Figure 2: The incision in the diagram is located in the upper half of the left flank, above the costochondral junction. In fact, it's identical to the Utrecht method. Does the incision position in the drawing correspond to that of sTPS? If not, can the drawing be modified?
Discussion
-Line 259: A citation appears with the number 46, but the paper only has 42 references.
Conclusions
-Lines 291 to 293: The need to improve the surgical device has not been previously discussed. Needle improvements are addressed and described in the Utrecht technique, using a threaded needle for each suture line. The surgeon inserts it from inside the abdominal cavity and the assistant completely removes it, along with the suture, from the outside. A straight needle or a straightened S-curve needle, 8 to 10 cm long, can be used. If a straight needle is used, a silicone plug can be placed on the blunt end to more easily pierce the abdominal wall. The needle is withdrawn from the outside by the assistant, and the surgeon retains the plug. (Nichols, S., & Fecteau, G. (2018). Surgical Management of Abomasal and Small Intestinal Disease. The Veterinary clinics of North America. Food animal practice, 34(1), 55–81. https://doi.org/10.1016/j.cvfa.2017.10.007)
References
-Lines 351-352: Reference 24 is not found in the text and does not appear to be related to the topic.
Author Response
The article describes the surgical treatment and subsequent follow-up of five cases of LDA in pregnant cows during the last month of pregnancy. Although similar clinical cases have been described in individual cows, to our knowledge, there are no clinical case series of LDA in cows at advanced stages of pregnancy. Furthermore, the proposed surgical technique will be useful for surgeons of small stature. In addition, the results are exceptionally good. For all this we congratulate the authors.
Response: Thank you for your high regard for our paper. The corrected parts in the revised version are highlighted with yellow boxes.
The authors justify the need for the new technique because being one-step laparoscopic abomasopexy is a highly effective method for correcting left displacement of the abomasum (LDA) in pregnant cows, it is not accessible to all veterinarians because of the need for expensive equipment and substantial experience.
In addition, the study argues that the toggle-pin suture (TPS) technique is also not appropriate for advanced pregnant cows due to the risk of uterine torsion.
Response: We agree completely with your comments, based on our experience.
Suggestion: The introduction and discussion focus on evaluating the efficacy of sTPS in relation to conventional TPS and laparoscopy-assisted TPS.
But, the most significant weakness observed in this work is that the technique recommended in all textbooks, and papers (Your reference number 15: Niehaus, A.J. Surgical management of abomasal disease) for LDA in pregnant cows is the left flank abomasopexy or Utrecht method. In the same books it is said that if the surgeon is small, the incision simply needs to be extended downward.
However, this technique is only mentioned in the introduction and is not addressed in the discussion. In our opinion, sTPS is a left-flank abomasopexy in which the incision is made more ventrally, to facilitate the work of smaller veterinarians, and in which the abomasopexy is performed with a toggle-pin suture instead of a conventional suture. Therefore, both the introduction and discussion should address left-flank abomasopexy in greater detail.
Response: According to this suggestion, new sentences are added in lines 274-279 of the revised version.
Suggestion: On the other hand, the advantages and disadvantages of the traditional abomasal suture used in the Utrecht method versus the toggle-pin suture used in this paper should be discussed.
Response: We think that the advantage of the surgical toggle-pin suture procedure is as follows: Suture-associated infections are unlikely to occur in our technique, in contrast to the problematic effect of the retained suture in Utrecht method. In the revised version, new sentences are added in lines 128-129 and 201-206. Additionally, by adding these sentences, one new reference paper is included.
Suggestion: -Line 41 and 42: Two-flank abomasopexy is not a common surgical technique, the 3 cited cases are of individual problematic animals with concomitant problems.
Response: According to this suggestion, the description “two-flank abomasopexy” is deleted in the revised version. Additionally, due to the correction, reference papers 9 and 14 were removed
Suggestion; -Line 42: Cite the original authors of the left flank abomasopexy (Lagerwey E, and Numans SR. De operatieve behandeling van de lebmaagdislocatie bij het rund volgens de ‘Utrechtse methode’. TijdschrDiergeneeskd 1968; 93: 366-76.) Since citation number 13 is not very appropriate for a paper as it is a workshop.
Response: According to this suggestion, the paper (Lagerweij and Numans, 1968) is added. Additionally, because the contents in reference paper 13 were used except for description of the left flank abomasopexy, this paper is kept as the reference paper.
Suggestion: -Lines 86 to 88: Given that Figure 1 does not show the entire cow, that the ventral area of the left flank is a very large area, and that the novelty of the technique is the incision site, can you specify exactly where the incision is located: how many cm caudal to rib 13, and whether the dorsal edge of the incision is at the level of the costochondral junction of said rib or is it located higher? A lateral photograph of only the patient with the incision made would be very illustrative.
Response: Unfortunately, we have no lateral photo showing the entire area of the left body. The incision is made at 2 to 3 cm caudal to the thirteenth rib. The middle of the incision is at the height of the costochondral junction of the rib. In the revised version, the description of the incision part is added in the lines 88-89.
Suggestion: -Lines 92 to 102: In many cases, the LDA is well above the incision line. In the Utrecht technique, abomasopexy is performed on the greater curvature of the abomasum, in the most dorsal part of it when it is displaced, but with a 15 cm rigid cannula and a low incision. How is the greater curvature accessed? Detail precisely in which parts of the abomasum the cannula is introduced.
Response: Fortunately, we could access the greater curvature through an incision made at the level of the costochondral junction of the 13th rib in all five cases. However, the greater curvature may not always be observed through this opening. Thus, this description is added in lines 286-289.
Suggestion: Lines 94-95: Explain the materials used to make the rod and suture.
Response: In the sentence, the term “self-made” was not accurate, A TPS device was made by the processing company (BS Medical Co.,Ltd.), according to our order to create this device. Thus, in the revised version, the term “self-made” is replaced by the company’s name.
Suggestion: -Lines 109 to 112: Determine the exact anatomical location of the fixation area on the patient's abdominal wall.
Response: The fixation part was the midline of the ventral abdominal wall cranial to the udder and caudal to the xiphoid process. This part is written in lines 116 of the revised version.
Suggestion: -Lines 119 to 121: Figure 1 B shows at least two muscular layers. But the text says "muscular layer." Can you specify which muscles and fascia are sutured and how?
Response: In this sentence, the term “muscular layer” is replaced by “muscular layers”. The muscular layers include the abdominal oblique and rectus muscles.
Suggestion: -Lines 119-122: Specify the suture material since neither Opepolix nor Suprylon are available on the Internet.
Response: The photo of this suture material is shown as follow:
Suggestion: -Line 136, Figure 2: The incision in the diagram is located in the upper half of the left flank, above the costochondral junction. In fact, it's identical to the Utrecht method. Does the incision position in the drawing correspond to that of sTPS? If not, can the drawing be modified?
Response: We completely agree with your suggestion that the location of the incision differed from the location written in the text in the diagram. Thus, the diagram is replaced by a modified version.
Suggestion: -Line 259: A citation appears with the number 46, but the paper only has 42 references.
Response: In this sentence, this number was mistaken. The correct number is corrected (Fodor’s report).
Suggestion: -Lines 291 to 293: The need to improve the surgical device has not been previously discussed. Needle improvements are addressed and described in the Utrecht technique, using a threaded needle for each suture line. The surgeon inserts it from inside the abdominal cavity and the assistant completely removes it, along with the suture, from the outside. A straight needle or a straightened S-curve needle, 8 to 10 cm long, can be used. If a straight needle is used, a silicone plug can be placed on the blunt end to more easily pierce the abdominal wall. The needle is withdrawn from the outside by the assistant, and the surgeon retains the plug. (Nichols, S., & Fecteau, G. (2018). Surgical Management of Abomasal and Small Intestinal Disease. The Veterinary clinics of North America. Food animal practice, 34(1), 55–81. https://doi.org/10.1016/j.cvfa.2017.10.007)
Response: We agree with your suggestion that the need to improve the surgical device has not been previously discussed in this sentence. This sentence is replaced by a new sentence in the revised version.
Suggestion: -Lines 351-352: Reference 24 is not found in the text and does not appear to be related to the topic.
Response: Thank you for pointing out this mistake. Reference 24 is removed.
Reviewer 3 Report (New Reviewer)
Comments and Suggestions for Authors
Main comments
This is an interesting short communication reporting the use of a toggle for correction of LDA using a left-sided abomasopexy approach, offering a compromise between traditional and laparoscopic approaches. The report is of interest, and the use of the toggle is novel; however this approach seems to me to be the same as the more widely used Utrecht approach with the sole difference being the use of the toggle rather than a needle and thus seems to offer few advantages. It is therefore a curious omission that the authors have not discussed this, and I think readers would want to know why this is the case (especially as the authors have briefly discussed how their approach compares to laparoscopic correction). My main suggestion for this manuscript would be that the authors include some discussion of the similarities between their technique and the well-established Utrecht approach (maybe suggest why they feel use of a toggle is preferable to a needle), as it currently feels like a significant oversight.
Minor comments
Line 37: It is better to avoid starting a sentence with an acronym, I would start this sentence with ‘left displaced abomasum’
Line 40: Replace ‘does’ with ‘do’ (after gravid uterus) for improved grammar because this refers to ‘mechanical effects’ which is plural
Lines 46-47: I am not sure I agree with this statement – rolling is contraindicated in pregnancy, and the roll and toggle approach is not recommended for LDA correction in pregnant animals by most authors. The reference you cite did include a small number of pregnant animals, but this is not reflective of recommended practice as you imply in lines 46-47. Furthermore, in the cited study all cows were operated on following rolling which means this approach was not performed in order to be minimally invasive, as suggested. I would recommend that you rephrase this as the recommended approach to LDA for pregnant cows is a left flank abomasopexy, which is what you have done – it is misleading to suggest that your approach is novel in this respect.
Line 74: Replace ‘is’ with ‘was’ (‘…in clinical sign in these animals is a sudden…’) for improved grammar
Lines 83-84: It would be good here to be clearer about your local anaesthetic block to aid reader understanding – did you perform what would be commonly referred to as a line block?
Line 146: This line would read better if it started ‘Mean gestation day was…’. Currently, the way it starts (‘In five cases…’) implies there are more cases you are not reporting and (incorrectly) suggests a lack of transparency.
Line 148: Specifying that five animals were sedated is unnecessary here as this is all of your study group (either just state ‘animals were sedated’ or ‘all animals were sedated’) – similar to above, specifying five suggests that only a subset were sedated but I don’t think this was the case?
Line 162: Correct the date here and the spelling of November – there are only 30 days in November so 31st cannot be correct.
Line 167: It would help the reader's understanding if ‘surgical time’ was defined – is this 1st incision to final suture or are you including prep time as well?
Author Response
Suggestion: This is an interesting short communication reporting the use of a toggle for correction of LDA using a left-sided abomasopexy approach, offering a compromise between traditional and laparoscopic approaches. The report is of interest, and the use of the toggle is novel; however this approach seems to me to be the same as the more widely used Utrecht approach with the sole difference being the use of the toggle rather than a needle and thus seems to offer few advantages. It is therefore a curious omission that the authors have not discussed this, and I think readers would want to know why this is the case (especially as the authors have briefly discussed how their approach compares to laparoscopic correction). My main suggestion for this manuscript would be that the authors include some discussion of the similarities between their technique and the well-established Utrecht approach (maybe suggest why they feel use of a toggle is preferable to a needle), as it currently feels like a significant oversight.
Response: Thank you for your kind suggestion. The corrected parts in the revised version are highlighted with yellow boxes. We agree with your suggestion that we have not discussed the technical comparison between sTPS and Utrecht approach. We think that there is the merit in using a TPS device in our surgical method. The Utrecht method requires the use of suture material to secure between the abomasum and thoracic walls. After cutting in Utrecht approach, the suture material is retained into the abdomen. The retained suture can sometimes be the source of infection, causing peritonitis. On the other hand, in sTPS, a TPS device can be picked up 1 to 2 days after cutting the suture; therefore, the suture is not retained within the abdomen. The description is added in lines 128-129 and 201-206. As another merit, in the procedure of sTPS, the cannula introduced in the abdominal cavity helps evacuate intraluminal gas and throw a TPS device into the abomasal lumen, as the two procedures can be performed almost at the same time. This can lead to a reduction in surgical time.
Suggestion: Line 37: It is better to avoid starting a sentence with an acronym, I would start this sentence with ‘left displaced abomasum’
Response: According to this suggestion, “LDA” is replaced by “left displaced abomasum”. Additionally, throughout the text, the sentences starting with acronyms are corrected.
Suggestion: Line 40: Replace ‘does’ with ‘do’ (after gravid uterus) for improved grammar because this refers to ‘mechanical effects’ which is plural
Response: According to this suggestion, this sentence is corrected.
Suggestion: Lines 46-47: I am not sure I agree with this statement – rolling is contraindicated in pregnancy, and the roll and toggle approach is not recommended for LDA correction in pregnant animals by most authors. The reference you cite did include a small number of pregnant animals, but this is not reflective of recommended practice as you imply in lines 46-47. Furthermore, in the cited study all cows were operated on following rolling which means this approach was not performed in order to be minimally invasive, as suggested. I would recommend that you rephrase this as the recommended approach to LDA for pregnant cows is a left flank abomasopexy, which is what you have done – it is misleading to suggest that your approach is novel in this respect.
Response: The reference papers (numbers 4,16, and 17) used for this manuscript included also paper created from reviewing previous reports, although they included papers showing results from a small number of pregnant animals. The description “as a minimally invasive procedure to correct the LDA” in line 49-50 (in previously submitted paper) is deleted, because it can lead to misunderstanding. We agree with your suggestion that sTPS would be the modified left flank abomasopexy. In the Introduction section, that is already noted (lines 65-67).
Suggestion: Line 74: Replace ‘is’ with ‘was’ (‘…in clinical sign in these animals is a sudden…’) for improved grammar.
Response: According to this suggestion, this sentence is corrected.
Suggestion: Lines 83-84: It would be good here to be clearer about your local anaesthetic block to aid reader understanding – did you perform what would be commonly referred to as a line block?
Response: The five cattle have been treated with infiltration anesthesia before surgery. In the revised version, this sentence is partly corrected.
Suggestion: Line 146: This line would read better if it started ‘Mean gestation day was…’. Currently, the way it starts (‘In five cases…’) implies there are more cases you are not reporting and (incorrectly) suggests a lack of transparency.
Response: According to this suggestion, this sentence is corrected.
Suggestion: Line 148: Specifying that five animals were sedated is unnecessary here as this is all of your study group (either just state ‘animals were sedated’ or ‘all animals were sedated’) – similar to above, specifying five suggests that only a subset were sedated but I don’t think this was the case?
Response: According to this suggestion, this sentence is deleted in the revised version.
Suggestion: Line 162: Correct the date here and the spelling of November – there are only 30 days in November so 31st cannot be correct.
Response: Thank you for pointing out our mistake. It was December, not November.
Suggestion: Line 167: It would help the reader's understanding if ‘surgical time’ was defined – is this 1st incision to final suture or are you including prep time as well?
Response: The surgical time means the time taken for the surgical process of Steps 1 to 4. In the revised version, this sentence is corrected.
Round 2
Reviewer 2 Report (New Reviewer)
Comments and Suggestions for Authors
Thank you for your kind responses and the corrections you made, which have cleared up all my doubts.
Congratulations on the paper.
This manuscript is a resubmission of an earlier submission. The following is a list of the peer review reports and author responses from that submission.
Round 1
Reviewer 1 Report
Comments and Suggestions for Authors
The manuscript aims to present a case-control study describing a surgical variant of the minimally invasive one-step laparoscopic abomasopexy procedure according to Christiansen. Based on the images provided, the procedure appears ergonomically uncomfortable to perform, as the surgeons are consistently bent over or squatting due to the highly ventral approach adopted by the technique.
Nevertheless, I can confirm that this technique, when used in pregnant animals—as this reviewer has personally applied, albeit never described in the literature, on approximately twenty occasions over the past 25 years, including in cows at term pregnancy—has proven consistently effective. This has been achieved with a modification involving the use of the toggle pin equipped with two sutures, originally designed for the Janowitz technique, in order to avoid the need for double abomasal puncture. In essence, the procedure constitutes a one-step surgery performed without endoscopic guidance, but under direct visual control by the surgeon.
While I believe that the publication of this manuscript could contribute to the broad array of surgical techniques available for the treatment of LDA, the sample size is extremely limited (5 cases vs. 4 cases) to justify a full-length article intended to validate the technique. Moreover, the manuscript fails to acknowledge the limitation posed by the small sample size.
I therefore recommend that the manuscript be revised and resubmitted as a short communication.
The choice to use a subjective appetite score—being the only clinical parameter assessed postoperatively—is inadequate. It would have been more informative to measure actual feed intake, as done in the study by Seeger et al. (2006), which compared 60 Janowitz procedures to 60 Dirksen procedures in LDA cows.
Regarding the use of xylazine, it can generally be omitted in Holstein-Friesian cattle when good local anesthesia is achieved. Although the manuscript appropriately discusses the limitations of xylazine use in pregnant animals, I am unable to provide a definitive judgment, as the degree of excitability of the animals used is not reported. I would suggest that sedation be administered only when strictly necessary, and not as a routine part of the surgical protocol.
Lines 320–351: Recovery and survival rate
The sample size is far too small to allow for meaningful comparisons with other validated techniques.
Author Response
For Reviewer 1
Thank you for your kind suggestion. We will address each of your suggestions. In the revised version, the yellow boxes highlight the corrected parts according to the two reviewers’ suggestions.
Suggestion: The manuscript aims to present a case-control study describing a surgical variant of the minimally invasive one-step laparoscopic abomasopexy procedure according to Christiansen. Based on the images provided, the procedure appears ergonomically uncomfortable to perform, as the surgeons are consistently bent over or squatting due to the highly ventral approach adopted by the technique.
Answer: We agree with your opinion about the ergonomically uncomfortable position of the surgeon in our surgical technique. Thus, in the revised version, a new sentence is added in lines 186-187.
Suggestion: Nevertheless, I can confirm that this technique, when used in pregnant animals—as this reviewer has personally applied, albeit never described in the literature, on approximately twenty occasions over the past 25 years, including in cows at term pregnancy—has proven consistently effective. This has been achieved with a modification involving the use of the toggle pin equipped with two sutures, originally designed for the Janowitz technique, in order to avoid the need for double abomasal puncture. In essence, the procedure constitutes a one-step surgery performed without endoscopic guidance, but under direct visual control by the surgeon.
While I believe that the publication of this manuscript could contribute to the broad array of surgical techniques available for the treatment of LDA, the sample size is extremely limited (5 cases vs. 4 cases) to justify a full-length article intended to validate the technique. Moreover, the manuscript fails to acknowledge the limitation posed by the small sample size.
I therefore recommend that the manuscript be revised and resubmitted as a short communication.
Answer: We agree that the sample size is limited in this paper (as only five cows were treated with our surgical technique), because displacement of the abomasum rarely occurs during pregnancy. According to your suggestion, the article type in this paper is changed from “article” to “(short) communication”. According to the change of the article type, the clinical data from the use of right flank omentopexy for pregnant cows with displaced abomasum are removed in the revised version.
Suggestion: The choice to use a subjective appetite score—being the only clinical parameter assessed postoperatively—is inadequate. It would have been more informative to measure actual feed intake, as done in the study by Seeger et al. (2006), which compared 60 Janowitz procedures to 60 Dirksen procedures in LDA cows.
Answer: Unfortunately, we have not measured actual feed intake similar to Seeger’s study, although we have used an appetite score categorized into four grades based on amounts of food intake (similar to Morkoç’s study). However, this appetite score is a subjective measurement method. Thus, in the revised version, the clinical data using the appetite score are removed. According to this change, descriptions in the discussion are greatly corrected.
Suggestion: Regarding the use of xylazine, it can generally be omitted in Holstein-Friesian cattle when good local anesthesia is achieved. Although the manuscript appropriately discusses the limitations of xylazine use in pregnant animals, I am unable to provide a definitive judgment, as the degree of excitability of the animals used is not reported. I would suggest that sedation be administered only when strictly necessary, and not as a routine part of the surgical protocol.
Answer: In our protocol for sTPS, xylazine was used to facilitate a smooth surgical procedure. However, xylazine should be avoided as much as possible in pregnant animals, because it may induce abortion. In lines 102-103 of the revised version, the sedative condition due to the use of xylazine is described. Additionally, in lines 150-153, new sentences are added, corresponding to your opinion.
Suggestion: Lines 320–351: Recovery and survival rate
The sample size is far too small to allow for meaningful comparisons with other validated techniques.
Answer: According to your suggestion, in the revised version, we deleted the comparison of sTPS data with the clinical data from the use of right flank omentopexy, because of the small sample size.
Reviewer 2 Report
Comments and Suggestions for Authors
Reviewer comments for manuscript ID animals-3668157 entitled ‘Utility of the Surgical Toggle-pin Suture Technique for Left Displacement of the Abomasum in Pregnant Cattle’
General comments
Left Displacement of abdomen is a common clinical condition in dairy cows and emergency surgical intervention is warranted to save the life of the cows. Advance pregnancy further complicates the condition and surgical techniques can further stress the animal and affect recovery as well as morbidity duration. Conducting a surgery in a recumbent cow is highly stressful for the affected cows and can lead to delayed recovery due to additional stress.
The researchers have evaluated anew technique to surgical treat LDA in cows in standing position using a modified surgical toggle pin suture techniques referred here as sTPS technique. As revealed in this study through the statistical analysis of data, reduced surgery time further reduced stress on the cows leading to lowered morbidity and faster recovery from surgery. It is a nice innovation that can be further modified for more precision and easy manoeuvrability for the surgeons.
The manuscript is written well though few errors are there in the usage of English that might be due to non-native speakers. The photographs and the schematic diagrams are impressive and help in easy comprehension of the techniques. The graphs and tables are able to fully depict the results of the study. The sample size is too small to draw wider conclusions , however being a clinical study, it will encourage further work on this techniques and the affection. Some of the references are too old, to refer them for revealing epidemiology of the disease. I have few queries that I need the authors to address before I recommend the publication of the manuscript.
Specific Comments
Lines 89-91: Use of Xylazine Hydrochloride is contraindicated in pregnant cows as it induces uterine contractions leading to abortion. Can you please clarify this method of sedation in pregnant cows in your study?
Lines 286-88 : What was the reason for removal of suture after 7 days, despite the advantages of a longer duration of sutures to stay? Please clarify.
Comments on the Quality of English LanguageEnglish language needs few corrections in the usage of the language especially in the discussion portion of the manuscript.
Author Response
For Reviewer 2
General comments
Left Displacement of abdomen is a common clinical condition in dairy cows and emergency surgical intervention is warranted to save the life of the cows. Advance pregnancy further complicates the condition and surgical techniques can further stress the animal and affect recovery as well as morbidity duration. Conducting a surgery in a recumbent cow is highly stressful for the affected cows and can lead to delayed recovery due to additional stress.
The researchers have evaluated anew technique to surgical treat LDA in cows in standing position using a modified surgical toggle pin suture techniques referred here as sTPS technique. As revealed in this study through the statistical analysis of data, reduced surgery time further reduced stress on the cows leading to lowered morbidity and faster recovery from surgery. It is a nice innovation that can be further modified for more precision and easy manoeuvrability for the surgeons.
Answer: Thank you for your valuable and insightful feedback. In the revised version, the article category in our paper is changed from “Article” to “Communication”, according to Reviewer 1’s suggestion. In this change, the data from the right flank omentopexy are deleted. Additionally, the results of the appetite score are removed. We will answer each of your suggestions. In the revised version, the yellow boxes highlight the corrected parts according to the two reviewers’ suggestions.
Suggestion: The manuscript is written well though few errors are there in the usage of English that might be due to non-native speakers. The photographs and the schematic diagrams are impressive and help in easy comprehension of the techniques. The graphs and tables are able to fully depict the results of the study. The sample size is too small to draw wider conclusions , however being a clinical study, it will encourage further work on this techniques and the affection. Some of the references are too old, to refer them for revealing epidemiology of the disease. I have few queries that I need the authors to address before I recommend the publication of the manuscript.
Answer: The English terms in the revised version is corrected by an English-proofreading company (Editage Co. Ltd). We had to use older reference papers because they include important information required for this paper, such as the prevalence of displaced abomasum in pregnant cows, the etiological role of the gravid uterus in the affection of displaced abomasum, and the applicability of surgical techniques to correct displaced abomasum in pregnant cows.
Specific Comments
Suggestion; Lines 89-91: Use of Xylazine Hydrochloride is contraindicated in pregnant cows as it induces uterine contractions leading to abortion. Can you please clarify this method of sedation in pregnant cows in your study?
Answer: A new reference paper is added in the revised version. The dose-associated risk of abortion was already described in pregnant cows treated with xylazine. Additionally, according to Reviewer 1’s suggestion, the discussion is extended in this part about why sedation might not always be required for sTPS, and how the proper administration of local anesthesia can replace the use of xylazine.
Suggestion: Lines 286-88 : What was the reason for removal of suture after 7 days, despite the advantages of a longer duration of sutures to stay? Please clarify.
Answer: Our timing to remove TPS fixation was decided based on our previous observation of a slaughterhouse specimen, in which a 7-day fixation led to good adhesion between the abomasum and ventral abdominal wall in another cow treated with a blind TPS technique, followed by sudden death soon after removing the TPS suture (Macroscopic view). However, we think the interval between suturing and its removal should be extended.
